# A20 Enhances the Expression of the Proto-Oncogene C-Myc by Downregulating TRAF6 Ubiquitination after ALV-A Infection

**DOI:** 10.3390/v14102210

**Published:** 2022-10-07

**Authors:** Xueyang Chen, Xingming Wang, Yuxin Yang, Chun Fang, Jing Liu, Xiongyan Liang, Yuying Yang

**Affiliations:** 1College of Animal Science, Yangtze University, No.88, Jingmi Road, Jingzhou 434025, China; 2College of Agriculture, Yangtze University, No.88, Jingmi Road, Jingzhou 434025, China

**Keywords:** ALV-A, chicken A20, TRAF6, STAT3, c-myc, tumorigenic mechanism

## Abstract

Hens infected with avian leukosis virus subgroup A (ALV-A) experience stunted growth, immunosuppression, and potentially, lymphoma development. According to past research, A20 can both promote and inhibit tumor growth. In this study, DF-1 cells were infected with ALV-A rHB2015012, and Gp85 expression was measured at various time points. A recombinant plasmid encoding the chicken A20 gene and short hairpin RNA targeting chicken A20 (A20-shRNA) was constructed and transfected into DF-1 cells to determine the effect on ALV-A replication. The potential signaling pathways of A20 were explored using bioinformatics prediction, co-immunoprecipitation, and other techniques. The results demonstrate that A20 and ALV-A promoted each other after ALV-A infection of DF-1 cells, upregulated A20, inhibited TRAF6 ubiquitination, and promoted STAT3 phosphorylation. The phosphorylated-STAT3 (p-STAT3) promoted the expression of proto-oncogene c-myc, which may lead to tumorigenesis. This study will help to further understand the tumorigenic process of ALV-A and provide a reference for preventing and controlling ALV.

## 1. Introduction

Avian leukosis (AL) is an infectious, neoplastic disease caused by the avian leukemia virus (ALV) [1]. The ALVs belong to the α retrovirus genus of the Retroviridae family [2]. ALVs are subdivided into 11 subgroups (A-K) based on the difference between the envelope protein Gp85 and the host [1,3] and the pathogenicity of each subtype varies. The common subgroups are A, B, J, and K. The primary host of ALV-A is laying hens, but the virus can also infect broiler chickens, causing slow growth, immunosuppression, tumorigenesis, and death in broilers and breeders after infection [4,5]. Over time, the host of ALV-A expanded from laying hens to broiler chickens and caused myeloma in addition to lymphoma. Moreover, the efficiency of chicken farms is affected, with the chicken and related industries suffering significant economic losses [6,7].

A20, also known as tumor necrosis factor alpha-inducible protein 3 (TNFAIP3), is a potent anti-inflammatory factor that prevents various human diseases and has important clinical and biological implications [8,9]. Biochemically, A20 is a ubiquitin-editing enzyme that possesses both E3 ubiquitin ligase and deubiquitinase (DUB) activities [9]. A20 comprises 790 amino acids, one amino-terminal ovarian tumor (OTU) domain, and seven carboxy-terminal zinc finger (ZnF) domains. The DUB activity of the OTU domain is mediated by Cys103 [10,11], while the ZnF4 domain has E3 ubiquitin ligase activity [12,13]. Multiple single-nucleotide polymorphisms (SNPs) in the A20 gene locus have been linked to various autoimmune and inflammatory diseases [14,15,16,17,18]. Moreover, abnormal A20 expression is associated with cancer development. For instance, in B lymphoma, A20 expression in diseased tissues is lower than in normal tissues due to inactivation, and it works through the NF-κB signaling pathway [19,20,21]. Although research results on A20 expression in liver cancer vary, they all support the notion that there is a difference between A20 expression in normal tissues and diseased tissues [22,23,24]. A20 can inhibit the development of tumor cells through NF-κB, RAC1, FAK signaling pathways, and even glycolytic metabolism [25]. In breast cancer, A20 is expressed at higher levels in lesions than in normal tissues [26,27]. A20 monoubiquitinates Snail1 to prevent GSK3 from phosphorylating Snail1. Moreover, A20 binds to SOCS3 and induces its degradation, which relieves STAT3 signaling inhibition and promotes breast cancer tumor growth and metastasis [25]. A20 upregulation alleviates LPS-induced activation of the NF-κB signaling pathway and generates inflammatory responses in avian chicken intestinal epithelial cells [28]. The role of A20 in avian tumorigenic diseases, however, is undocumented. C-myc plays an important role in cell proliferation, growth, survival, and differentiation, and when its expression is disordered, can cause tumorigenesis [29,30].

Tumor-associated protein A20 and the proto-oncogene c-myc were upregulated in the liver transcriptome of 40-week-old laying hens artificially infected with HB2015012, suggesting that A20 may be involved in ALV-A infection or tumorigenesis. Therefore, in this study, A20 was overexpressed and knocked down, and the possible signaling pathway was verified to explore the tumorigenic mechanism of ALV-A mediated by A20. The findings facilitate the understanding of tumorigenesis mechanisms.

## 2. Materials and Methods

### 2.1. Cells, Viruses, and Plasmids

DF-1 cells (GNO30, Shanghai Cell Bank of Chinese Academy of Sciences) were cultured in Dulbecco’s Modified Eagle Medium (DMEM, C11995500, Gibco, USA) and supplemented with 10% fetal bovine serum (FBS, Gemini, 900-108, USA), 100 µg/mL streptomycin and 100 U/mL penicillin at 37 °C in 5% CO_2_. Virus strain rHB2015012 (GeneBank ID: KY612442) was rescued and preserved by our laboratory along with the plasmids, psi-flag, pCMV-HA, and pLKO.1-GFP.

### 2.2. Overexpression and shRNA Interference, Plasmid Construction, and Expression Analysis

The overexpression primers were designed based on the published NCBI sequences for chicken A20 (XR_005848478), TRAF6 (XM_040673313), and STAT3 (NM_001030931). A20 was amplified by PCR and cloned into the vectors, psi-flag, and pCMV-HA. Two shRNAs were designed (http://rnaidesigner.thermofisher.com/rnaiexpress/, accessed on 30 September 2022) and cloned into the vector PLKL.1-GFP after annealing. The primer sequences are shown in Table 1.

DF-1 cells were transfected with correctly identified overexpression plasmids and shRNA interference plasmids using TurboFect Transfection Reagent (R0531, Thermo Fisher, Waltham, MA, USA). After 24 h of transfection, cells were harvested and subjected to Western blotting to detect overexpression and interference effects. The transfection procedure was performed following the guidelines of the manufacturer.

### 2.3. RNA Extraction and Semi-Quantitative PCR

Total RNA was extracted using an RNA extraction kit (Tsingke, Wuhan, Beijing, China), and reverse transcription was performed using HiScripIII 1st Strand cDNA Synthesis Kit (Vazyme, Nanjing, China). The transfection procedure was carried out according to the instructions of the manufacturer. To amplify the reverse transcription product, 2 × Taq Plus Master Mix II (Vazyme, Nanjing, China) was used in semi-quantitative PCR. Image J and data normalization (sample gray value/β-actin gray value) were used for grayscale analysis. The primers for semi-quantitative PCR are shown in Table 2. The PCR reaction system consisted of 2 × PCR Mix 12.5 μL, forward primer 1 μL, reverse primer 1 μL, and cDNA 1.5 μL H_2_O 9 μL. The PCR reaction program consisted of 94 °C 3 min; 94 °C 30 s, 60 °C 30 s, 72 °C 30 s (total 25 cycle); 72 °C 5 min; and 4 °C keep warm.

### 2.4. Co-Immunoprecipitation Analysis

Co-immunoprecipitation and immunoblot analyses were performed as previously described [31]. Briefly, the expression plasmids, psi-flag-A20, pCMV-HA-A20, psi-flag-TRAF6, pCMV-HA-TRAF6, psi-flag-STAT3, and pCMV-HA-STAT3, were transfected into DF-1 cells prepared in advance. After the transfection, a freshly prepared DMEM medium containing 10 % FBS was replaced and cultured for 48 h. The cells were collected and lysed using NP-40 (P0013F, Beyotime, China), and an appropriate amount of protein A/G (Beyotime, Shanghai, China) was added. Anti-A20 (Servicebio, Wuhan, China), anti-TRAF6 (Bioss, Beijing, China), and anti-STAT3 (Bioss, Beijing, China) antibodies completed the co-immunoprecipitation experiment according to the manufacturer’s instructions.

### 2.5. Ubiquitination Assay

To analyze the effect of ALV-A (rHB015012) on TRAF6 ubiquitination in DF-1 cells, DF-1 cells were transfected with plasmids expressing HA-TRAF6 and Flag-Myc-Ub. After 24 h of transfection, cells were inoculated with 10^4^ 50% tissue culture infection volume (TCID_50_) rHB015012. Whole-cell extracts were immunoprecipitated with anti-Flag antibody, followed by Western blot analysis with anti-ALV-A Gp85 (prepared in our laboratory, 1:1000), anti-GAPDH (Proteintech, Wuhan, China 1:5000), and anti-A20 (Servicebio, Wuhan, China, 1:1000) antibodies. To analyze the effect of A20 on TRAF6 ubiquitination in DF-1 cells, DF-1 cells were transfected with plasmids expressing Flag-A20/shRNA-A20, HA-TRAF6, and Flag-Myc-Ub. After 24 h of transfection, cells were inoculated with 104 TCID50 rHB015012, harvested after 48 h, and whole-cell extracts were immunoprecipitated with anti-HA antibody, followed by Western blot analysis with anti-ALV-A Gp85, anti-GAPDH, anti-myc, and anti-A20 antibodies.

### 2.6. Western Blotting

Western blotting was performed as previously described [32]. The antibodies and dilutions used for Western blotting were as follows: anti-ALV-A envelope glycoprotein specific antibody (prepared in our laboratory, 1:1000), rabbit anti-TRAF6 antibody (bs-2830R, Bioss, China, 1:1000), rabbit anti-STAT3 antibody (bs-1141R, Bioss, China, 1:1000), rabbit anti-phospho-STAT3 (Ser727) antibody (bs-3429R, Bioss, China, 1:1000), rabbit anti-A20 antibody (GB112182, Servicebio, China, 1:1000), mouse anti-GAPDH monoclonal antibody (60004, Proteintech, USA, 1:5000), mouse anti-HA monoclonal antibody (M180, MBL, Japan, 1:3000), mouse anti-Myc monoclonal antibody (M047, MBL, Japan, 1:3000), mouse anti-Flag monoclonal antibody (Sangon Biotech, Shanghai, China, 1:5000), HRP-conjugated goat anti-rabbit IgG (Sangon Biotech, Shanghai, China, 1:5000), and HRP-conjugated goat anti-mouse IgG (Sangon Biotech, Shanghai, China, 1:5000).

### 2.7. Indirect Immunofluorescence Assay (IFA)

DF-1 cells transfected with plasmids psi-flag, psi-flag-STAT3, pLKO.1-GFP, and pLKO.1-STAT3 were cultured to 80% confluent in a 24-well plate and infected with or without ALV-A (rHB2015012) at a dose of 10^3^ TCID_50_. Cell culture was discarded at 72 h post-infection, cells were fixed with 4% paraformaldehyde for 10 min at room temperature (RT) after washing twice with PBS, and then cells were perforated with 0.5% Triton X-100 for 10 min at RT after washing with PBS for three times. The plate was blocked with 10% FBS at room temperature for 1 h, then incubated with anti-ALV-A Gp85 antibody (prepared in our laboratory, 1:1000) at appropriate dilution at RT for 45 min. After washing, FITC-conjugated goat anti-mouse IgG (Sangon Biotech, Shanghai, China) was incubated at RT for 1 h. Finally, the plate was washed three times with PBS, and images were taken by the fluorescence microscope (Leica DM i8 manual).

### 2.8. Statistical Analysis

GraphPad Prism 6 (https://www.graphpad.com/, accessed on 30 September 2022) was used for the statistical analysis of data. Data were presented as means ± standard error of the mean (SEM). The Student’s *t*-test analyzed the differences in data. *p* value of <0.05 was considered statistically significant (* *p* ≤ 0.05, ** *p* ≤ 0.01, and *** *p* ≤ 0.001).

## 3. Results

### 3.1. ALV-A Upregulates the Expression of A20

DF-1 cells were seeded in a 6-well plate at 37 °C with 5% CO_2_. After 24 h, 10^4^ TCID_50_ rHB2015012 viruses were added to each well. Cells were harvested three, four, and five days after infection. The transcription and expression of A20 were detected by semi-quantitative PCR and Western blotting. At 3, 4, and 5 dpi, the expression of A20 was greater in rHB2015012-infected cells than in negative cells. (Figure 1A,B).

### 3.2. Overexpression and Knockdown of A20 Could Influence ALV-A Virus Replication

To explore A20 function in ALV-A replication, we overexpressed and knocked down A20 in DF-1 cells. As Gp85 is a viral envelope glycoprotein with subgroup specificity, Western blotting was used to assess the A20 effect on ALV-A using Gp85. Western blotting results reveal that the overexpression of A20 in DF-1 cells promoted the replication of ALV-A. In contrast, the knockdown of A20 inhibited ALV-A replication (Figure 2A,B). As a group-specific antigen of ALV, p27 can provide a good assessment of virus replication; semi-quantitative PCR results are consistent with Western blotting (Figure 2C,D).

### 3.3. ALV-A Inhibits Ubiquitination of TRAF6 via A20

TRAF6 is important in the inflammatory response [33,34], and its overexpression in certain tumors regulates tumorigenesis and angiogenesis [35]. The expression of TRAF6 in DF-1 cells infected with ALV-A was detected by Western blotting. The results indicate that ALV-A upregulates TRAF6 by inhibiting its ubiquitination at 72 h of infection (Figure 3A,B).

To explore the association between A20 and TRAF6 in ALV-A, first by co-immunoprecipitation, we found that A20 and TRAF6 can interact (Figure 3C,D). Then, we overexpressed and knocked down A20 to verify its effect on TRAF6 expression. The results indicate that A20 overexpression upregulated TRAF6 by inhibiting its ubiquitination at 48 h of ALV-A infection (Figure 3E,G). On the other hand, knocking down A20 had the opposite effect of A20 overexpression, which promoted the ubiquitination of TRAF6 at 48 h of ALV-A infection (Figure 3F,H).

### 3.4. A20 Affects the Phosphorylation of STAT3 via TRAF6

In previous studies, the JAK-STAT signaling pathway, closely related to leukemia and myeloid production, was activated after ALV infection [36,37]. Among the seven STATs with known functions, STAT3 plays a key role in the malignant transformation of cells. First, we found that p-STAT3 was upregulated in DF-1 cells infected with ALV-A for 72 h (Figure 4A). At 48 h of ALV-A infection, p-STAT3 was upregulated after A20 overexpression (Figure 4D) and downregulated after A20 knockdown (Figure 4E). 

To explore the association between TRAF6 and STAT3, first, co-immunoprecipitation determined that TRAF6 and STAT3 interacted (Figure 4B,C). Then, we overexpressed and knocked down TRAF6 to assess its effect on p-STAT3. The results demonstrate that upregulation of TRAF6 expression promoted the upregulation of p-STAT3 at 48 h of ALV-A infection (Figure 4F,G).

### 3.5. STAT3 Promotes the Expression of C-Myc

C-myc is a classic oncogene that plays an important role in the occurrence and development of tumors. More than 70% of tumors have c-myc mutations or changes in expression [29,30]. The relative expression levels of tumor-related cytokines, Y box-binding protein-1 (YB-1), cyclin D1, p53, B-cell lymphoma-2 (Bcl-2), Cystatin A (CSTA), and c-myc were determined after ALV-A infection. It was found that c-myc was significantly upregulated, while cyclin D1 was downregulated at 48 h of ALV-A infection (Figure 5A). To determine whether STAT3 was responsible for the downregulation of cyclin D1, p53 and the upregulation of c-myc, CSTA, overexpression, and knockdown of STAT3 were carried out, respectively. By semi-quantitative PCR, it was found that the upregulation of c-myc was induced by STAT3 (Figure 5B,C). At the same time, the virus rHB2015012 was responsible for the downregulation of cyclin D1, p53, and upregulation of CSTA (Figure 5D–J). The IFA results show that DF-1 cells infected with ALV-A exhibited green fluorescence, while uninfected DF-1 cells showed no fluorescence (Figure 6).

## 4. Discussion

ALVs are one of the main culprits of avian tumor disease. There are currently no drugs or vaccines on the market, which complicates the protection of poultry resources in China. In 2015, a strain of ALV-A that can cause both lymphoma and myeloma was isolated from laying hens, and the tumorigenic type of this strain was verified through animal experiments [38]. To exclude interference from other viruses, the strain HB2015012 was rescued by reverse genetics to obtain a clearer background strain rHB2015012, which can still induce lymphocytoma and myeloma in laying hens. Our previous study found that the transcription of A20 was upregulated after ALV-A infection (HB2015012) in laying hens, suggesting that A20 may be involved in ALV-A infection or tumorigenesis. Moreover, the JAK-STAT signaling pathway [37], which is closely related to leukemia and myeloid production, was activated, which may be one of the reasons why ALV-A (HB2015012) caused myeloma.

Although many researchers have studied the relationship between A20 and tumors, the research on avian diseases is limited to the observation that the upregulation of A20 alleviates the LPS-induced activation of the NF-κB signaling pathway and the inflammatory response in chicken intestinal epithelial cells [28]. A20 plays different roles in different tumor types, acting as a promoter and a suppressor [25]. The expression of A20 was upregulated in DF-1 cells infected with ALV-A. A20 also promoted the replication of ALV-A, as confirmed by A20 overexpression and knockdown. Uncertainty remains as to whether the two can perform this function indefinitely or whether there is a compensatory mechanism. Time and severity of onset are directly proportional to viral load in the body [39]. A20 can inhibit the activation of the NF-κB signaling pathway by inhibiting TRAF6 ubiquitination [40]. The same results were obtained in this study. A20 interacted with TRAF6 and inhibited its ubiquitination, resulting in its upregulation after ALV-A infection. A positive correlation was found between the overexpression and knockdown of A20. Recent findings suggest that TRAF6 regulates tumorigenesis by inhibiting apoptosis and stimulating the proliferation and invasion of various cancers [41,42,43]. TRAF6 is upregulated and downregulated in tumors. When TRAF6 is overexpressed, it confers at least three known cancer hallmarks, including maintenance of proliferative signaling, resistance to cell death, and induction of angiogenesis [42], in pancreatic [44] and breast [45] cancers. When downregulated, it inhibits cell proliferation, invasion, and migration and promotes cell apoptosis [42], for instance, in MCF-7 breast cancer cell lines and SCCHN cells [45,46].

The JAK/STAT signaling pathway is widely present in various tissues and cells. Moreover, its excessive activation is closely associated with the occurrence, development, invasion, and metastasis of multiple tumors [47,48]; STAT3 enhances pro-survival signaling necessary for T-cell proliferation. However, if the JAK/STAT pathway is inadequately regulated, it may contribute to T-cell lymphoma development [37]. We found that the phosphorylation level of STAT3 was increased in DF-1 cells infected with ALV-A. To determine whether the increase in p-STAT3 is associated with A20 and TRAF6, we first overexpressed and knocked down A20, followed by TRAF6. Both overexpression and knockdown displayed that TRAF6 positively regulated STAT3 phosphorylation following ALV-A infection.

STAT3 regulates the transcription of cell cycle regulatory proteins and proto-oncogenes [49], including B-cell lymphoma BCL-2, BCL-6, BCL-xL, c-myc, and cyclin D1, among others. As ALV-A infection promotes cell proliferation and delays cell death, the proliferation regulators c-myc and cyclin D1 were detected. The results demonstrate that c-myc was regulated by STAT3 after ALV-A infection, while cyclin D1, p53, and CSTA were only related to ALV-A and not STAT3. Confusingly, STAT3-regulated Bcl-2, a proto-oncogene associated with lymphocytomas, did not show differential expression after ALV-A rHB2015012 infection, nor did it show differential expression after STAT3 overexpression and interference, possibly because of the incompatibility of the cells used in the study or to the virus itself. Interestingly, proliferation experiments showed that the overexpression of c-myc could promote the proliferation of DF-1 cells and delay apoptosis. This may be due to the function of proto-oncogene c-myc promoting cell carcinogenesis.

In conclusion, our results show that ALV-A (HB2015012) infection increased the expression level of A20 and inhibited the ubiquitination of TRAF6, resulting in its upregulation. This upregulation promoted the STAT3 phosphorylation, which stimulated the proto-oncogene c-myc expression, leading to tumorigenesis (Figure 7). The results of this study can help identify molecular targets for treating and preventing ALV.

## Figures and Tables

**Figure 1 viruses-14-02210-f001:**
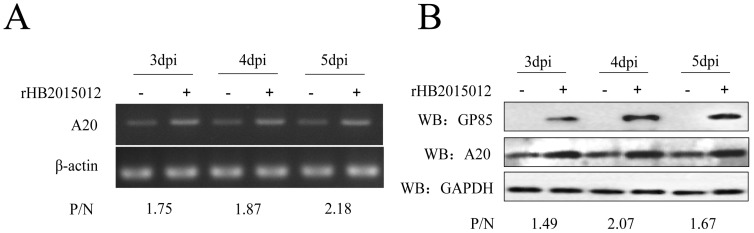
ALV-A (rHB2015012) upregulates A20 expression. (**A**) Detecting ALV-A promoted A20 expression using semi-quantitative PCR. (**B**) Detecting ALV-A promoted A20 expression using Western blotting.

**Figure 2 viruses-14-02210-f002:**
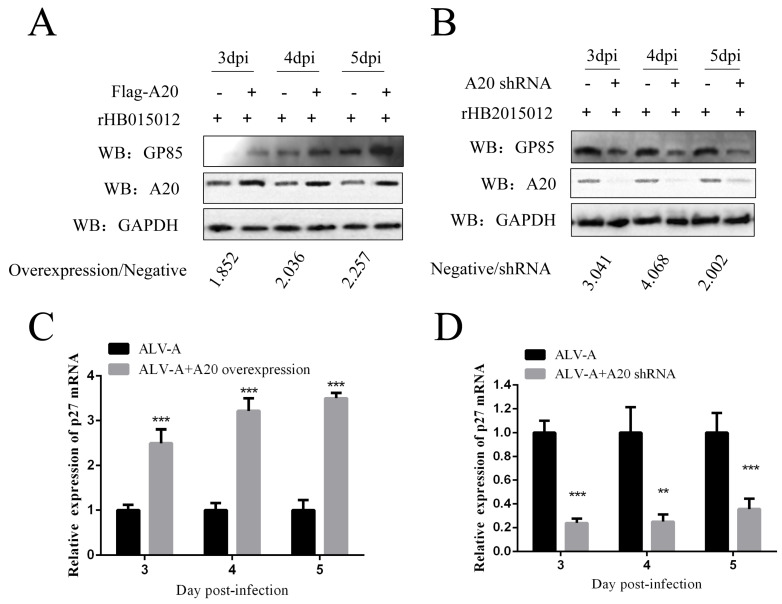
A20 promotes replication of ALV-A (rHB2015012) (n = 3 and three tests in total). (**A**,**C**) A20 overexpression promotes replication of ALV-A rHB2015012. (**B**,**D**) A20 knockdown inhibits replication of ALV-A rHB2015012. ** *p* value ≤ 0.01, and *** *p* value ≤ 0.001. “+”, Indicates the addition of the plasmid or virus. “−”, Indicates that the plasmid or virus has not been added.

**Figure 3 viruses-14-02210-f003:**
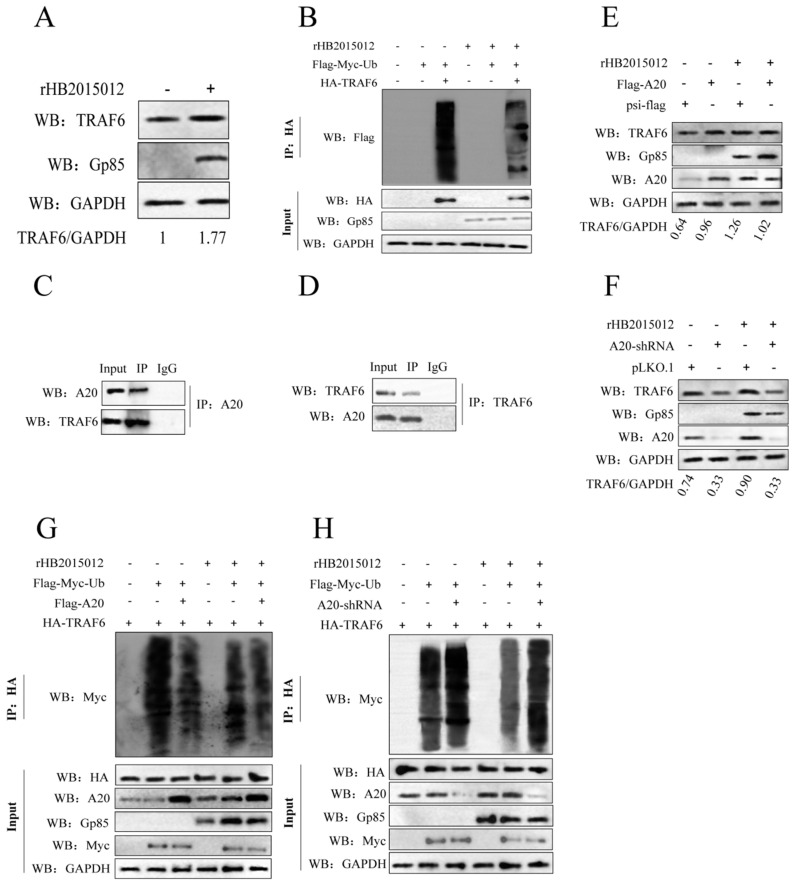
A20 upregulates TRAF6 expression by inhibiting TRAF6 ubiquitination after ALV-A infection. (**A**) The expression of TRAF6 was upregulated at 72 h of ALV-A infection. (**B**) CO-IP analysis of the effect of ALV-A (rHB2015012) on TRAF6 ubiquitination in DF-1 cells using the expression plasmids Flag-Myc-Ub and HA-TRAF6. (**C**,**D**) In vitro analysis of the interaction between A20 and TRAF6 in DF-1 cells transfected with plasmids expressing Flag-A20 and HA-TRAF6 (**C**) or HA-A20 and Flag-TRAF6 (**D**). IP, immunoprecipitation. (**E**,**F**) A20 overexpression and interference affected the expression of TRAF6 at 48 h of ALV-A infection. (**G**,**H**) The effect of A20 on TRAF6 ubiquitination in DF-1 cells was analyzed using the expression plasmids Flag-Myc-Ub, HA-TRAF6, Flag-A20, and A20-shRNA by CO-IP analysis after ALV-A (rHB2015012) infection. “+”, Indicates the addition of the plasmid or virus. “−”, Indicates that the plasmid or virus has not been added.

**Figure 4 viruses-14-02210-f004:**
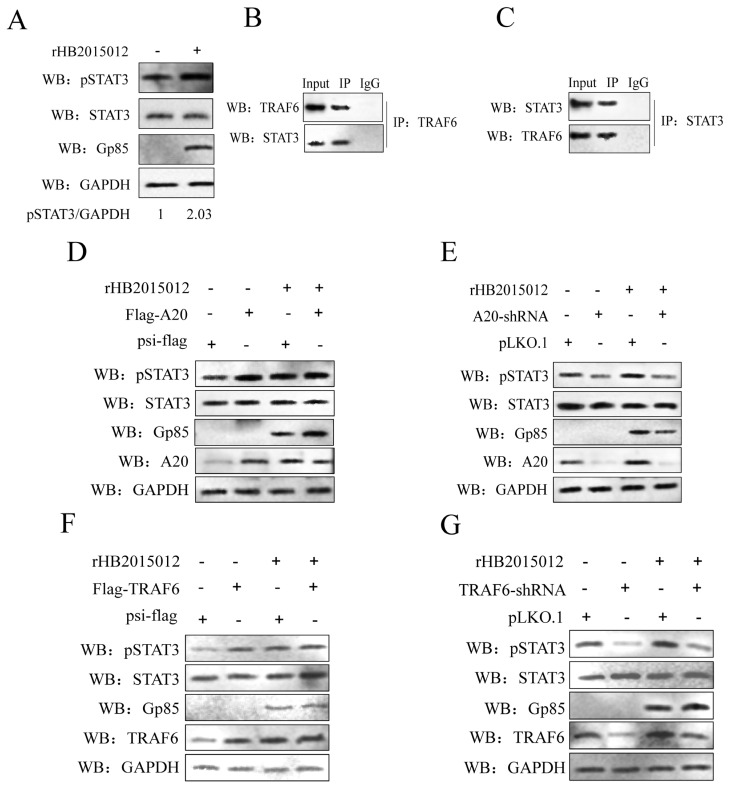
Upregulated TRAF6 after ALV-A infection promotes phosphorylation of STAT3. (**A**) p-STAT3 is upregulated at 72 h of ALV-A infection. (**B**,**C**) In vitro analysis of the interaction between TRAF6 and STAT3 in DF-1 cells transfected with plasmids expressing Flag-TFAF6 and HA-STAT3 (**B**) or HA-TRAF6 and Flag-STAT3 (**C**). IP, immunoprecipitation. (**D**,**E**) A20 overexpression and interference affected p-STAT3 expression at 48 h of ALV-A infection. (**F**,**G**) TRAF6 overexpression and interference affected the expression of p-STAT3 at 48 h of ALV-A infection.

**Figure 5 viruses-14-02210-f005:**
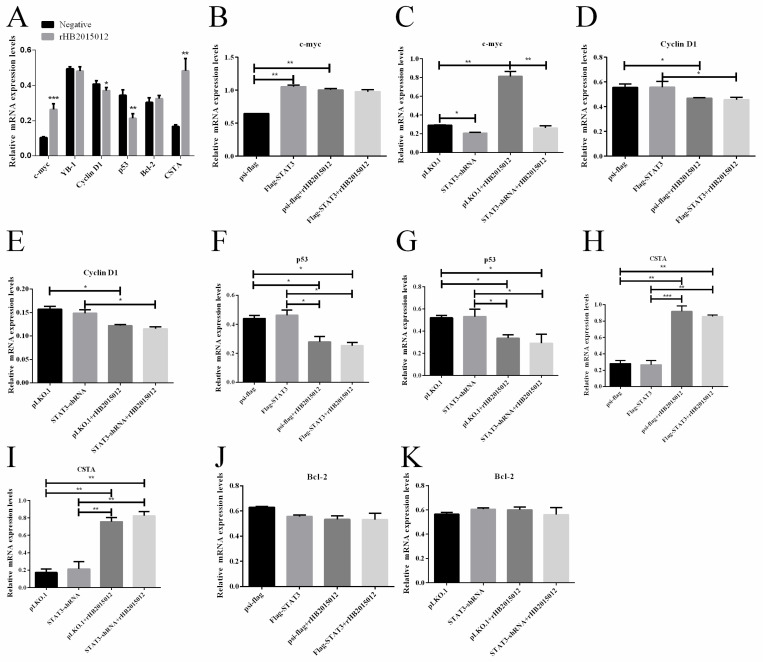
Excessive transcription of c-myc regulated by STAT3 after ALV-A infection (n = 3 and three tests in total). (**A**) ALV-A promoted c-myc expression, inhibited cyclin D1, and did not affect YB-1 at 48 h of infection. (**B**,**C**) STAT3 promotes c-myc expression at 48 h of ALV-A infection. (**D**,**E**) Cyclin D1 expression inhibition at 48 h of ALV-A infection but has nothing to do with STAT3. (**F**,**G**) Inhibition of p53 expression at 48 h of ALV-A infection but has nothing to do with STAT3. (**H**,**I**) CSTA expression was promoted at 48 h of ALV-A infection, but it has nothing to do with STAT3. (**J**,**K**) STAT3 overexpression and interference did not affect YB-1 expression. * *p* value ≤ 0.05, ** *p* value ≤ 0.01, and *** *p* value ≤ 0.001.

**Figure 6 viruses-14-02210-f006:**
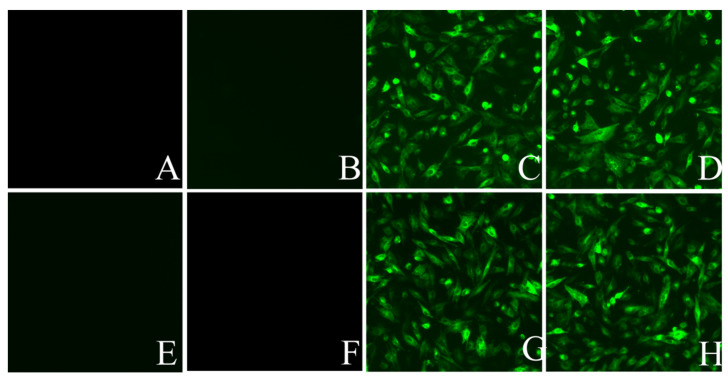
Indirect immunofluorescence assay (400×), (**A**) DF-1 cells transfected with psi-flag plasmid. (**B**) DF-1 cells transfected with psi-flag-STAT3 plasmid. (**C**) DF-1 cells transfected with psi-flag plasmid and infection with ALV-A rHB2015012. (**D**) DF-1 cells transfected with psi-flag-STAT3 plasmid and infection with ALV-A rHB2015012. (**E**) DF-1 cells transfected with pLKO.1-GFP plasmid. (**F**) DF-1 cells transfected with pLKO.1-STAT3 plasmid cells. (**G**) DF-1 cells transfected with pLKO.1-GFP plasmid and infection with ALV-A rHB2015012. (**H**) DF-1 cells transfected with pLKO.1-STAT3 plasmid and infection with ALV-A rHB2015012.

**Figure 7 viruses-14-02210-f007:**
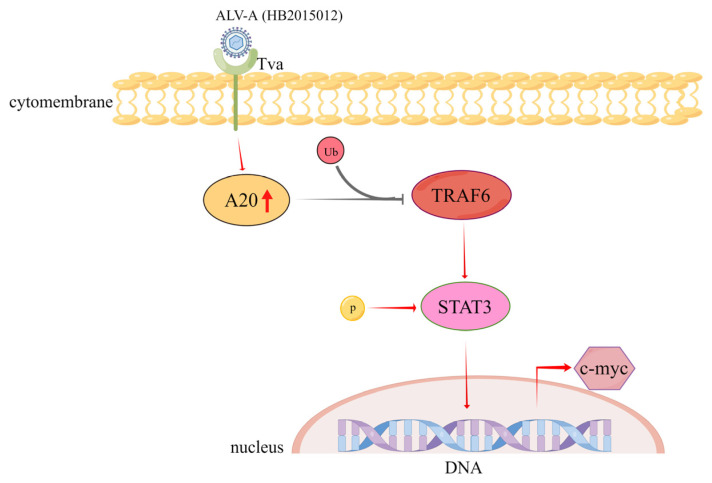
ALV-A (rHB2015012) promotes tumorigenesis signaling through A20. ALV-A (rHB2015012) infection increases the expression level of A20. It inhibits TRAF6 ubiquitination, leading to its upregulation, and TRAF6 upregulation promotes STAT3 phosphorylation, which in turn promotes the proto-oncogene c-myc expression, resulting in tumorigenesis.

**Table 1 viruses-14-02210-t001:** Primers for plasmid construction.

Plasmids	Primers	Sequences (5′-3′)
psi-flag-A20	F	CGCGGATCCATGGCTGGCCAACACATCCTTCCTC
	R	CCGCTCGAGGCCGTAGATCTGTTTGAACTGGTAGCATTCA
pCMV-HA-A20	F	GCCATGGAGGCCCGAATTCGGTCGACCATGGCTGGCCAACACATCCT
	R	ATCCCCGCGGCCGCGGTACCTCGAGGCCGTAGATCTGTTTGAACTGGT
psi-flag-TRAF6	F	CGGGATCCATGAGCTTGCTACACAGTGATAG
	R	CCGCTCGAGTTACGCAGCTCCATCAGTACTG
pCMV-HA-TRAF6	F	CCATGGAGGCCCGAATTCGGTCGACCATGAGCTTGCTACACAGTGATAGC
	R	ATCCCCGCGGCCGCGGTACCTCGAGTTACGCAGCTCCATCAGTACTG
psi-flag-STAT3	F	CGGGATCCATGGCGCAGTGGAACCAACT
	R	CCGCTCGAGTCACATTGGTGAGGAAGCACACT
pCMV-HA-STAT3	F	CCATGGAGGCCCGAATTCGGTCGACCATGGCGCAGTGGAACCAACT
	R	ATCCCCGCGGCCGCGGTACCTCGAGTCACATTGGTGAGGAAGCACACT
pLKO.1-GFP-A20	F	GATCCGCTTTGTATCAGAGCAATATGTTCAAGAGACATATTGCTCTGATACAAAGCTTTTTTG
	R	AATTCAAAAAAGCTTTGTATCAGAGCAATATGTCTCTTGAACATATTGCTCTGATACAAAGCG
pLKO.1-GFP-TRAF6	F	GATCCGCAGCAGATGCCTAACCATTATTCAAGAGATAATGGTTAGGCATCTGCTGCTTTTTTG
	R	AATTCAAAAAAGCAGCAGATGCCTAACCATTATCTCTTGAATAATGGTTAGGCATCTGCTGCG
pLKO.1-GFP-STAT3	F	GATCCGCTGTCAGCCATGGAGTATGTTTCAAGAGAACATACTCCATGGCTGACAGCTTTTTTG
	R	AATTCAAAAAAGCTGTCAGCCATGGAGTATGTTCTCTTGAAACATACTCCATGGCTGACAGCG
psi-flag-myc-chUb	F	GGGCGGATCCGCGATACCGGAATTCGAGCAGAAGCTGATCTCAGAGGAGGACCTGATGCAGATCTTCGTGAAGACCCTG
	R	TGCGCCTGAGGGGAGGCTAACTCGAGCAATTCGTCGAGGGACCTAA

**Table 2 viruses-14-02210-t002:** Primers for semi-quantitative PCR.

Gene	Primers	Sequences (5′-3′)
YB-1	F	ACCGTGGAGTTTGATGTGGTT
	R	CTTCCGGGATGTTCTCTGCTC
Cyclin D1	F	TCAAGTGCGTGCAGAAGGAA
	R	CTGCGGTCAGAGGAATCGTT
c-myc	F	TTCTTTGCCCTGCGTGACC
	R	GCCTCAACTGCTCTTTCTCTGC
β-actin	F	CAACACACTGCTGTCTGGTGGTA
	R	ATCGTACTCCTGCTTGCTGATCC
A20	F	TGGGGCTCGAAACAGACTTC
	R	TTGTCGTAGCCGAGCACAAT
P27	F	CCGGGGGAATTGGTTGCTAT
	R	ATCTGGCTGTGACTTCTGCC
Bcl2	F	CGCTACCAGAGGGACTTCG
	R	TTGACCCCATCACGGAAGAG
CSTA	F	ACCGGACTCAAGTGGTTGC
	R	CCGGTAAGGCTGGGATGTT
P53	F	ATGGCGGAGGAGATGGAACC
	R	CAATGGCAGAGGTGGTGGTG

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
