# Peer review of "A20 Enhances the Expression of the Proto-Oncogene C-Myc by Downregulating TRAF6 Ubiquitination after ALV-A Infection"

_viruses, 2022, doi:10.3390/v14102210_

Round 1
Reviewer 1 Report
The manuscript reports the role of A20 on regulating the expression of oncogenic genes after ALV-A infection and its signal pathway, it has a good significance for understanding the mechanism of ALV causing myeloma. However, the MS needs major revision, the major suggestions are following:
1. In Introduction, the statement on ALV-A oncogenic species and oncogenic genes should be added.
2. In backgroud introduction, the authors mentioned in their previous studies that ALV-A infection cause the increase of A20 expression. It is unknown whether the increase of A20 is in the tumor tissues of infected chickens or in the other tissues without tumor? Is there any detection of oncogenic genes such as myc? It is suggested to increase the detection of some oncogenic genes to incease the logic of the design.
3. In methods and results, ALV-A detection such as viral titers or IFA should be added; the detections of other related oncogenic genes should be added.
4. Dose the proliferation or cycle of DF1 cells after the increase of myc gene expression change? Please add statement in the MS.
5. The photograph of ubiquitination results is not clear, and it is difficult to distinguish whether the ubiquitination occurs or not? It is suggested to replace with a better one.
6. For the Co-IP graphes, more detailed descriptions should be added in the notes or result descriptions.
7. Different ALV-A strains cause different tumorigenic types, tumorigenic time, virus replication speed, and even some strains contain some tumorigenic genes; It is unknown whether the ALV-A strain used in this MS has been verified on these characteristics? If yes, they should be added in discussions.
Author Response
The manuscript reports the role of A20 on regulating the expression of oncogenic genes after ALV-A infection and its signal pathway, it has a good significance for understanding the mechanism of ALV causing myeloma. However, the MS needs major revision, the major suggestions are following:
Response: We appreciate the reviewer’s critical comments and suggestions! The manuscript was carefully modified according to the reviewers’ comments and suggestions, and changes were highlighted in the revised version.
Point 1: In Introduction, the statement on ALV-A oncogenic species and oncogenic genes should be added.
Response: We thanks for the reviewer’s suggestion! Added description of tumorigenic types of ALV-A and proto-oncogene c-myc in Introduction.
Point 2: In backgroud introduction, the authors mentioned in their previous studies that ALV-A infection cause the increase of A20 expression. It is unknown whether the increase of A20 is in the tumor tissues of infected chickens or in the other tissues without tumor? Is there any detection of oncogenic genes such as myc? It is suggested to increase the detection of some oncogenic genes to incease the logic of the design.
Response: Sorry for making the reviewer confusing! Tumor-associated protein A20 and proto-oncogene c-myc were up-regulated in the liver transcriptome of 40-week-old laying hens artificially infected with HB2015012. In addition, activation of the JAK-STAT signaling pathway associated with myeloma was also found. Since the data is unpublished, it is briefly described here.
Point 3: In methods and results, ALV-A detection such as viral titers or IFA should be added; the detections of other related oncogenic genes should be added.
Response: We thanks for the reviewer’s suggestion! In the revised version, we added description about the IFA in the methods and Result. Added detections for other oncogenes to the results.
Point 4: Dose the proliferation or cycle of DF1 cells after the increase of myc gene expression change? Please add statement in the MS.
Response: Thanks! Proliferation experiments were performed on DF-1 cells after STAT3 overexpression using cck-8 kit, the results showed that it can promote cell proliferation and delay apoptosis. Added this result description to the Discussion.
Point 5: The photograph of ubiquitination results is not clear, and it is difficult to distinguish whether the ubiquitination occurs or not? It is suggested to replace with a better one.
Response: The ubiquitination test was performed by transfecting the overexpression plasmids HA-TRAF6 and Flag-Myc-Ub in DF-1 cells, and detected by co-IP. Finally, gray values were analysed using the software Image J to determine the amount of ubiquitination. If it is further confirmed that K48 or K63 is the best, further testing is abandoned due to limited conditions. With regard to the picture, it is not that I do not wish to exchange it for a better, which, under our present conditions, is the best.
Point 6: For the Co-IP graphes, more detailed descriptions should be added in the notes or result descriptions.
Response: Thanks very much! We thanks for the reviewer’s suggestion! In the revised version, we rewrote the figure legends and highlighted the changes.
Point 7: Different ALV-A strains cause different tumorigenic types, tumorigenic time, virus replication speed, and even some strains contain some tumorigenic genes; It is unknown whether the ALV-A strain used in this MS has been verified on these characteristics? If yes, they should be added in discussions.
Response: We thanks for the reviewer’s suggestion! In 2015, a strain of ALV-A that can cause both lymphoma and myeloma was isolated from laying hens, and the tumorigenic type of this strain was verified through animal experiments[38]. In order to exclude the interference of other viruses, the strain HB2015012 was rescued by reverse genetics to obtain a more clear background strain rHB2015012, which can still induce lymphocytoma and myeloma in laying hens.

Reviewer 2 Report
The current study reveals a tumorigenesis mechanism induced by ALV-A. A20 downregulated the ubiquitination of TRAF6, causing c-myc expression enhancement. This study will help to further understand the tumorigenic process of ALV-A.
There are some issues listed below that would be critical to address.
The way this current manuscript is written, makes it difficult for a reader (or reviewer) to understand what has been done exactly. The language must be improved.
Figure legends and methods are the two worst parts of the manuscript. They lack consistency and are impossible to follow – let alone if one wants to replicate the authors experiments.
Authors may need to add some description about figure 3, 4 and 5 in legends as well as in main text, which are too briefly.
In Figure 1, why did the authors choose to analyze A20 expression from 3 dpi and not 1 dpi?
In Figure 2, gp85 was used to evaluate the replication of ALV-A. However, gp85 is just one of viral structural elements of ALV-A, authors should include an assay to prove the viral replication (like qRT-PCR or other experiments to prove viable virus output) if they want to make the statement that in infected cells significantly increased.
In Figure 3A, the authors did not mention the time point at which TRAF6 expression was detected. It should be consistent with the time point at which A20 expression was detected.
From Figure 3 to 5, all results and figure legends have no description of the time-points of experiments.
For co-immunoprecipitation experiments (both in the main paper and supplemental), negative controls are lacking. It requires an IgG isotype negative control for the IP.
Although the authors suggest A20 involves the phosphorylation of STAT3 via TRAF6 and STAT3 promotes c-myc expression, it does not imply downregulating the ubiquitination of TRAF6 is the only pathway through which A20 promotes c-myc expression. Authors should add some rescue experiments to prove it. The expression levels of related factors should also be verified in vivo.
Author Response
The current study reveals a tumorigenesis mechanism induced by ALV-A. A20 downregulated the ubiquitination of TRAF6, causing c-myc expression enhancement. This study will help to further understand the tumorigenic process of ALV-A. There are some issues listed below that would be critical to address.
Response: We thanks the reviewer’s critical comments and suggestions very much! We have revised the manuscript was carefully according to the reviewers’ comments and suggestions, and changes were highlighted in the revised version.
Point 1: The way this current manuscript is written, makes it difficult for a reader (or reviewer) to understand what has been done exactly. The language must be improved.
Response: We are sorry for these errors that trouble the reviewer. We have carefully proofread the manuscript and sent the manuscript for English editing. We hope these errors are minimized.
Point 2: Figure legends and methods are the two worst parts of the manuscript. They lack consistency and are impossible to follow – let alone if one wants to replicate the authors experiments.
Response: Sorry for making the reviewer confusing! In the revised version, we rewrote the figure legends and methods that highlighted the changes.
Point 3: Authors may need to add some description about figure 3, 4 and 5 in legends as well as in main text, which are too briefly.
Response: Thanks! In the revised version, we rewrote the figure legends and highlighted the changes.
Point 4: In Figure 1, why did the authors choose to analyze A20 expression from 3 dpi and not 1 dpi?
Response: A20 was not significantly up-regulated within two days after ALV-A rHB2015012 infection, and significantly increased at 3dpi, so we chose to start from the 3dpi.
Point 5: In Figure 2, gp85 was used to evaluate the replication of ALV-A. However, gp85 is just one of viral structural elements of ALV-A, authors should include an assay to prove the viral replication (like qRT-PCR or other experiments to prove viable virus output) if they want to make the statement that in infected cells significantly increased.
Response: Thanks! p27 as a group-specific antigen of ALV can be a good assessment of virus replication, adding transcriptional analysis of p27 using semi-quantitative PCR. The result is shown in Figures 2C–D.
Point 6: In Figure 3A, the authors did not mention the time point at which TRAF6 expression was detected. It should be consistent with the time point at which A20 expression was detected.
Response: Sorry! We have revised the errors and marked them in the revised manuscript.
Point 7: From Figure 3 to 5, all results and figure legends have no description of the time-points of experiments.
Response: Sorry! We have added time points to the Methods, Results and section figure legends.
Point 8: For co-immunoprecipitation experiments (both in the main paper and supplemental), negative controls are lacking. It requires an IgG isotype negative control for the IP.
Response: Thanks very much! We rework the IP experiments and modify Figures 3C-D and 4B-C. First, we performed a pre-experiment before the formal IP experiment, with almost no background interference. Second,since the protein molecular weight of Flag-A20 (96 KD) , HA-A20 (92 KD), Flag-TRAF6 (66 KD), HA-TRAF6 (62 KD), Flag-STAT3 (91 KD), HA-STAT3 (87 KD) is quite different from the relative protein molecular weight of the antibody, it can be distinguished. In addition, to avoid the interference of IgG, we use the method of reducing the power supply voltage and increasing the electrophoresis time in SDS-PAGE, so that the bands can be separated as much as possible.
Point 9: Although the authors suggest A20 involves the phosphorylation of STAT3 via TRAF6 and STAT3 promotes c-myc expression, it does not imply downregulating the ubiquitination of TRAF6 is the only pathway through which A20 promotes c-myc expression. Authors should add some rescue experiments to prove it. The expression levels of related factors should also be verified in vivo.
Response: Thanks very much! In the first place, I fully agree with you that you are correct. There is more than one pathway leading to overexpression of C-myc. Such as, p53 mutations can lead to overexpression of c-myc. For other reasons, more people are needed to study. In this study, we focused on how A20 overexpresses c-myc. This experiment was carried out on the basis of the previous transcriptome (the liver transcriptome of rHB2015012-infected laying hens). Although the expression levels of the relevant factors were different in vivo (relative expression fold), the results were consistent.

Round 2
Reviewer 2 Report
Although the authors have improved the manuscript, there are still some issues. The suggestions are as follows:
1. In figure 2C and D, why were the data from the control group not subjected to error analysis? The number of samples in the experiment and the number of repetitions of the experiment should also be clearly written in figure legend. This should also be done in figure 5.
2. Indirect immunofluorescence assays are added as in figure 6, but the relative results are not described in the manuscript.
3. In line 70 and 153, what is the meaning of CO2? CO2 should be changed to CO2.
4. In addition to western blot experiments, the dilution of the antibody and the source of the antibody should be added to the description of other experiments, such as co-immunoprecipitation analysis, ubiquitination assay and IFA assay.
5. There are still many minor mistakes in the manuscript, and authors should check the manuscript more carefully.
Author Response
1. In figure 2C and D, why were the data from the control group not subjected to error analysis? The number of samples in the experiment and the number of repetitions of the experiment should also be clearly written in figure legend. This should also be done in figure 5.
Response: We re-analyzed Fig.2C and D and made changes to the pictures. We have added the number of samples and experiments to the legend of Figures 2 and 5.
2. Indirect immunofluorescence assays are added as in figure 6, but the relative results are not described in the manuscript.
Response: Thanks! We have added a description about IFA results in the manuscript.
3. In line 70 and 153, what is the meaning of CO2? CO2 should be changed to CO2.
Response: Thanks! Change has made in the revised version.
4. In addition to western blot experiments, the dilution of the antibody and the source of the antibody should be added to the description of other experiments, such as co-immunoprecipitation analysis, ubiquitination assay and IFA assay.
Response: Thanks! Change has made in the revised version.
5. There are still many minor mistakes in the manuscript, and authors should check the manuscript more carefully.
Response: We appreciate the reviewer’s critical comments and suggestions! The manuscript was carefully modified according to the reviewers’ comments and suggestions, and changes were highlighted in the revised version.
